# Legendre Memory Units: Continuous-Time Representation in Recurrent Neural Networks

**Aaron R. Voelker**[1,2]        **Ivana Kajić**[1]        **Chris Eliasmith**[1,2]

[1]Centre for Theoretical Neuroscience, Waterloo, ON     [2]Applied Brain Research, Inc.
{arvoelke, i2kajic, celiasmith}@uwaterloo.ca

## Abstract

We propose a novel memory cell for recurrent neural networks that dynamically maintains information across long windows of time using relatively few resources. The Legendre Memory Unit (LMU) is mathematically derived to orthogonalize its continuous-time history – doing so by solving $d$ coupled ordinary differential equations (ODEs), whose phase space linearly maps onto sliding windows of time via the Legendre polynomials up to degree $d - 1$. Backpropagation across LMUs outperforms equivalently-sized LSTMs on a chaotic time-series prediction task, improves memory capacity by two orders of magnitude, and significantly reduces training and inference times. LMUs can efficiently handle temporal dependencies spanning 100,000 time-steps, converge rapidly, and use few internal state-variables to learn complex functions spanning long windows of time – exceeding state-of-the-art performance among RNNs on permuted sequential MNIST. These results are due to the network's disposition to learn scale-invariant features independently of step size. Backpropagation through the ODE solver allows each layer to adapt its internal time-step, enabling the network to learn task-relevant time-scales. We demonstrate that LMU memory cells can be implemented using $m$ recurrently-connected Poisson spiking neurons, $\mathcal{O}(m)$ time and memory, with error scaling as $\mathcal{O}(d/\sqrt{m})$. We discuss implementations of LMUs on analog and digital neuromorphic hardware.

## 1 Introduction

A variety of recurrent neural network (RNN) architectures have been used for tasks that require learning long-range temporal dependencies, including machine translation [3, 26, 34], image caption generation [36, 39], and speech recognition [10, 16]. An architecture that has been especially successful in modelling complex temporal relationships is the LSTM [18], which owes its superior performance to a combination of memory cells and gating mechanisms that maintain and nonlinearly mix information over time.

LSTMs are designed to help alleviate the issue of vanishing and exploding gradients commonly associated with training RNNs [5]. However, they are still prone to unstable gradients and saturation effects for sequences of length $T > 100$ [2, 22]. To combat this problem, extensive hyperparameter searches, gradient clipping strategies, layer normalization, and many other RNN training "tricks" are commonly employed [21].

Although standard LSTMs with saturating units have recently been found to have a memory of about $T = 500$–1,000 time-steps [25], non-saturating units in RNNs can improve gradient flow and scale to 2,000–5,000 time-steps before encountering instabilities [7, 25]. However, signals in realistic natural environments are continuous in time, and it is unclear how existing RNNs can cope with conditions

as $T \rightarrow \infty$. This is particularly relevant for models that must leverage long-range dependencies within an ongoing stream of continuous-time data, and run in real time given limited memory.

Interestingly, biological nervous systems naturally come equipped with mechanisms that allow them to solve problems relating to the processing of continuous-time information – both from a learning and representational perspective. Neurons in the brain transmit information using spikes, and filter those spikes continuously over time through synaptic connections. A spiking neural network called the Delay Network [38] embraces these mechanisms to approximate an ideal delay line by converting it into a finite number of ODEs integrated over time. This model reproduces properties of "time cells" observed in the hippocampus, striatum, and cortex [13, 38], and has been deployed on ultra low-power [6] analog and digital neuromorphic hardware including Braindrop [28] and Loihi [12, 37].

This paper applies the memory model from [38] to the domain of deep learning. In particular, we propose the Legendre Memory Unit (LMU), a new recurrent architecture and method of weight initialization that provides theoretical guarantees for learning long-range dependencies, even as the discrete time-step, $\Delta t$, approaches zero. This enables the gradient to flow across the continuous history of internal feature representations. We compare the efficiency and accuracy of this approach to state-of-the-art results on a number of benchmarks designed to stress-test the ability of recurrent architectures to learn temporal relationships spanning long intervals of time.

## 2   Legendre Memory Unit

**Memory Cell Dynamics**   The main component of the Legendre Memory Unit (LMU) is a memory cell that orthogonalizes the continuous-time history of its input signal, $u(t) \in \mathbb{R}$, across a sliding window of length $\theta \in \mathbb{R}_{>0}$. The cell is derived from the linear transfer function for a continuous-time delay, $F(s) = e^{-\theta s}$, which is best-approximated by $d$ coupled ordinary differential equations (ODEs):

$$\theta \dot{\mathbf{m}}(t) = \mathbf{A}\mathbf{m}(t) + \mathbf{B}u(t) \tag{1}$$

where $\mathbf{m}(t) \in \mathbb{R}^d$ is a state-vector with $d$ dimensions. The ideal state-space matrices, $(\mathbf{A}, \mathbf{B})$, are derived through the use of Padé [30] approximants [37]:

$$\mathbf{A} = [a]_{ij} \in \mathbb{R}^{d \times d}, \quad a_{ij} = (2i+1) \begin{cases} -1 & i < j \\ (-1)^{i-j+1} & i \geq j \end{cases} \tag{2}$$

$$\mathbf{B} = [b]_i \in \mathbb{R}^{d \times 1}, \quad b_i = (2i+1)(-1)^i, \quad i, j \in [0, d-1].$$

The key property of this dynamical system is that $\mathbf{m}$ represents sliding windows of $u$ via the Legendre [24] polynomials up to degree $d-1$:

$$u(t - \theta') \approx \sum_{i=0}^{d-1} \mathcal{P}_i \left( \frac{\theta'}{\theta} \right) m_i(t), \quad 0 \leq \theta' \leq \theta, \quad \mathcal{P}_i(r) = (-1)^i \sum_{j=0}^{i} \binom{i}{j} \binom{i+j}{j} (-r)^j \tag{3}$$

where $\mathcal{P}_i(r)$ is the $i^{\text{th}}$ shifted Legendre polynomial [32]. This gives a unique and optimal decomposition, wherein functions of $\mathbf{m}$ correspond to computations across windows of length $\theta$, projected onto $d$ orthogonal basis functions.

**Discretization**   We map these equations onto the memory of a recurrent neural network, $\mathbf{m}_t \in \mathbb{R}^d$, given some input $u_t \in \mathbb{R}$, indexed at discrete moments in time, $t \in \mathbb{N}$:

$$\mathbf{m}_t = \bar{\mathbf{A}} \mathbf{m}_{t-1} + \bar{\mathbf{B}} u_t \tag{4}$$

where $(\bar{\mathbf{A}}, \bar{\mathbf{B}})$ are the discretized matrices provided by the ODE solver for some time-step $\Delta t$ relative to the window length $\theta$. For instance, Euler's method supposes $\Delta t$ is sufficiently small:

$$\bar{\mathbf{A}} = (\Delta t / \theta)\, \mathbf{A} + \mathbf{I}, \quad \bar{\mathbf{B}} = (\Delta t / \theta)\, \mathbf{B}. \tag{5}$$

We also consider discretization methods such as zero-order hold (ZOH) as well as those that can adapt their internal time-steps [9].

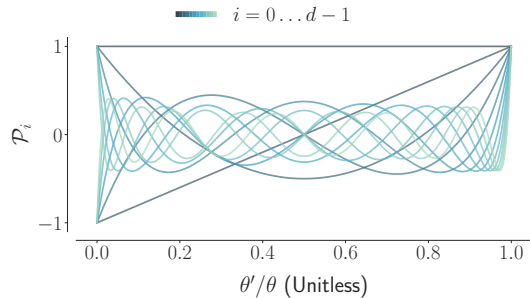

Figure 1: Shifted Legendre polynomials ($d = 12$). The memory of the LMU represents the entire sliding window of input history as a linear combination of these scale-invariant polynomials. Increasing the number of dimensions supports the storage of higher-frequency inputs relative to the time-scale.

**Approximation Error**    When $d = 1$, the memory is analogous to a single-unit LSTM without any gating mechanisms (i.e., a leaky integrator with time-constant $\theta$). As $d$ increases, so does its memory capacity relative to frequency content. In particular, the approximation error in equation 3 scales as $\mathcal{O}\left(\theta\omega/d\right)$, where $\omega$ is the frequency of the input $u$ that is to be committed to memory [38].

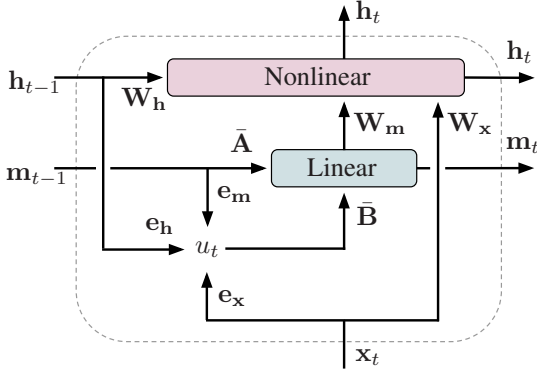

Figure 2: Time-unrolled LMU layer.    An $n$-dimensional state-vector ($\mathbf{h}_t$) is dynamically coupled with a $d$-dimensional memory vector ($\mathbf{m}_t$). The memory represents a sliding window of $u_t$, projected onto the first $d$ Legendre polynomials.

**Layer Design**    The LMU takes an input vector, $\mathbf{x}_t$, and generates a hidden state, $\mathbf{h}_t \in \mathbb{R}^n$. Each layer maintains its own hidden state and memory vector. The state mutually interacts with the memory, $\mathbf{m}_t \in \mathbb{R}^d$, in order to compute nonlinear functions across time, while dynamically writing to memory. Similar to the NRU [7], the state is a function of the input, previous state, and current memory:

$$\mathbf{h}_t = f\left(\mathbf{W_x}\mathbf{x}_t + \mathbf{W_h}\mathbf{h}_{t-1} + \mathbf{W_m}\mathbf{m}_t\right) \quad (6)$$

where $f$ is some chosen nonlinearity (e.g., $\tanh$) and $\mathbf{W_x}$, $\mathbf{W_h}$, $\mathbf{W_m}$ are learned kernels. Note this decouples the size of the layer's hidden state ($n$) from the size of the layer's memory ($d$), and requires holding $n + d$ variables in memory between time-steps. The input signal that writes to the memory (via equation 4) is:

$$u_t = \mathbf{e_x}^\mathsf{T}\mathbf{x}_t + \mathbf{e_h}^\mathsf{T}\mathbf{h}_{t-1} + \mathbf{e_m}^\mathsf{T}\mathbf{m}_{t-1} \quad (7)$$

where $\mathbf{e_x}$, $\mathbf{e_h}$, $\mathbf{e_m}$ are learned encoding vectors. Intuitively, the kernels ($\mathbf{W}$) learn to compute nonlinear functions across the memory, while the encoders ($\mathbf{e}$) learn to project the relevant information into the memory. The parameters of the memory ($\bar{\mathbf{A}}$, $\bar{\mathbf{B}}$, $\theta$) may be trained to adapt their time-scales by backpropagating through the ODE solver [9], although we do not require this in our experiments.

This is the simplest design that we found to perform well across all tasks explored below, but variants in the form of gating $u_t$, forgetting $\mathbf{m}_t$, and bias terms may also be considered for more challenging tasks. Our focus here is to demonstrate the advantages of learning the coupling between an optimal linear dynamical memory and a nonlinear function.

## 3  Experiments

Tasks were selected with the goals of validating the LMU's derivation while succinctly highlighting its key advantages: it can learn temporal dependencies spanning $T = 100{,}000$ time-steps, converge rapidly due to the use of non-saturating memory units, and use relatively few internal state-variables to compute nonlinear functions across long windows of time. The source code for the LMU and our experiments are published on GitHub.[1]

Proper weight initialization is central to the performance of the LMU, as the architecture is indeed a specific way of configuring a more general RNN in order to learn across continuous-time representations. Equation 2 and ZOH are used to initialize the weights of the memory cell ($\bar{\mathbf{A}}$, $\bar{\mathbf{B}}$). The time-scale $\theta$ is initialized based on prior knowledge of the task. We find that $\bar{\mathbf{A}}$, $\bar{\mathbf{B}}$, $\theta$ do not require training for these tasks since they can be appropriately initialized. We recommend equation 3 as an option to initialize $\mathbf{W_m}$ that can improve training times. The memory's feedback encoders are initialized to $\mathbf{e_m} = \mathbf{0}$ to ensure stability. Remaining kernels ($\mathbf{W_x}$, $\mathbf{W_h}$) are initialized to Xavier normal [15] and the remaining encoders ($\mathbf{e_x}$, $\mathbf{e_h}$) are initialized to LeCun uniform [23]. The activation function $f$ is set to $\tanh$. All models are implemented with Keras and the TensorFlow backend [1] and run on CPUs and GPUs. We use the Adam optimizer [20] with default hyperparameters, monitor the validation loss to save the best model, and train until convergence or 500 epochs. We note that our method does not require layer normalization, gradient clipping, or other regularization techniques.

## 3.1 Capacity Task

The copy memory task [18] is a synthetic task that stress-tests the ability of an RNN to store a fixed amount of data—typically 10 values—and persist them for a long interval of time. We consider a variant of this task that we dub the capacity task. It is designed to test the network's ability to maintain $T$ values in memory for large values of $T$ relative to the size of the network. We do so in a controlled way by setting $T = 1/\Delta t$ and then scaling $\Delta t \to 0$ while keeping the underlying data distribution fixed. Specifically, we randomly sample from a continuous-time white noise process, band-limited to $\omega = 10\,\mathrm{Hz}$. Each sequence iterates through 2.5 seconds of this process with a time-step of $\Delta t = 1/T$. At each step, the network must recall the input values from $\lfloor iT/(k-1)\rfloor$ time-steps ago for $i \in \{0, 1, \ldots k-1\}$. Thus, the task evaluates the network's ability to maintain $k$ points along a sliding window of length $T$, using only its internal state-variables to persist information between time-steps. We compare an LSTM to a simplified LMU, for $k = 5$, while scaling $T$.

**Isolating the Memory Cell** To validate the function of the LMU memory in isolation, we disable the kernels $\mathbf{W_x}$ and $\mathbf{W_h}$, as well as the encoders $\mathbf{e_h}$ and $\mathbf{e_m}$, set $\mathbf{e_x} = 1$, and set the hidden activation $f$ to the identity. We use $d = 100$ dimensions for the memory. This simplifies the architecture to 500 parameters that connect a linear memory cell to a linear output layer. This is done to demonstrate that the LMU is initially disposed to remember sequences of length $\theta$ (set to $T$ steps).

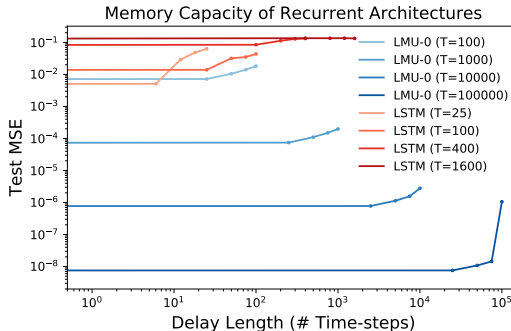

**Model Complexity** We compare the isolated LMU memory cell to an LSTM with 100 units connected to a 5-dimensional linear output layer. This model contains ~41k parameters, and 200 internal state-variables—100 for the hidden state, and 100 for the "carry" state—that can be leveraged to maintain information between time-steps. Thus, this task is theoretically trivial in terms of internal memory storage for $T \leq 200$. The LMU has $n = 5$ hidden units and $d = 100$ dimensions for the memory (equation 4). Thus, the LMU is using significantly fewer computational resources than the LSTM, 500 vs 41k parameters, and 105 vs 200 state-variables.

Figure 3: Comparing LSTMs to LMUs while scaling the number of time-steps between the input and output. Each curve corresponds to a model trained at a different window length, $T$, and evaluated at 5 different delay lengths across the window. The LMU successfully persists information across $10^5$ time-steps using only 105 internal state-variables, and without any training. It is able to do so by maintaining a compressed representation of the 10 Hz band-limited input signal processed with a time-step of $\Delta t = 1/T$.

**Results and Discussion** Figure 3 summarizes the test results of each model, trained at different values of $T$, by reporting the MSE for each of the $k$ outputs separately. We find that the LSTM can solve this task when $T < 400$, but struggles for $T \geq 400$ due to the lack of hyperparameter optimization, consistent with [22, 25]. The LMU solves this task near-perfectly, since $\theta\omega = 10 \ll d$ [38]. In fact, the LMU does not even need to be trained for this task; testing is performed on the initial state of the network, without any training data. LMU performance continues to improve with training (not shown), but that is not required for the task. We note that performance improves as $T \to \infty$ because this yields discretized numerics that more closely follow the continuous-time descriptions of equations 1 and 3. The next task demonstrates that the representation of the memory generalizes to internally-generated sequences, that are not described by band-limited white noise processes, and are learned rapidly through mutual interactions with the hidden units.

## 3.2 Permuted Sequential MNIST

The permuted sequential MNIST (psMNIST) digit classification task [22] is commonly used to assess the ability of RNN models to learn complex temporal relationships [2, 7, 8, 21, 25]. Each $28 \times 28$ image is flattened into a one-dimensional pixel array and permuted by a fixed permutation matrix. Elements of the array are then provided to the network one pixel at a time. Permutation distorts the

temporal structure in the image sequence, resulting in a task that is significantly more difficult than the unpermuted version.

**State-of-the-art**  Current state-of-the-art results on psMNIST for RNNs include Zoneout [21] with 95.9% test accuracy, indRNN [25] with 96.0%, and the Dilated RNN [8] with 96.1%. One must be careful when comparing across studies, as each tend to use different permutation seeds, which can impact the overall difficulty of the task. More importantly, to allow for a fair comparison of computational resources utilized by models, it is necessary to consider the number of state-variables that must be modified in memory as the input is streamed online during inference. In particular, if a network has access to more than $28^2 = 784$ variables to store information between time-steps, then there is very little point in attempting this task with an RNN [4, 7]. That is, it becomes trivial to store all 784 pixels in a buffer, and then apply a feed-forward network to achieve state-of-the-art. For example, the Dilated RNN uses an internal memory of size $50 \cdot (2^9 - 1) \approx 25k$ (i.e., 30x greater than 784), due to the geometric progression of dilated connections that must buffer hidden states in order to skip them in time. Parameter counts for RNNs are ultimately poor measures of resource efficiency if a solution has write-access to more internal memory than there are elements in the input sequence.

**LMU Model**  Our model uses $n = 212$ hidden units and $d = 256$ dimensions for the memory, thus maintaining $n + d = 468$ variables in memory between time-steps. The hidden state is projected to an output softmax layer. This is equivalent to the NRU [7] in terms of state-variables, and similar in computational resources, while the LMU has ~102k trainable parameters compared to ~165k for the NRU. We set $\theta = 784$ s with $\Delta t = 1$ s, and initialize $\mathbf{e_h} = \mathbf{e_m} = \mathbf{W_x} = \mathbf{W_h} = \mathbf{0}$ to test the ability of the network to learn these parameters. Training is stopped after 10 epochs, as we observe that validation loss is already minimized by this point.

**Results**  Table 1 is reproduced from Chandar et al. [7] with the following adjustments. First, the EURNN (94.50%) has been removed since it uses 1024 state-variables. Second, we have added the phased LSTM [29] with matched parameter counts and $\alpha = 10^{-4}$. Third, a feed-forward baseline is included, which simply projects the flattened input sequence to a softmax output layer. This informs us of how well we should expect a model to perform (92.65%) supposing it linearly memorizes the input and projects it to the output softmax layer. For the LMU and feed-forward baseline, we extended the code from Chandar et al. [7] in order to ensure that the training, validation, and test data were identical with the same permutation seed and batch size. All other results in Table 1 use ~165k parameters (LMU uses ~102k, and FF-baseline uses ~8k).

Table 1: Validation and test set accuracy for psMNIST (extended from [7])

| Model | Validation | Test |
|---|---|---|
| RNN-orth | 88.70 | 89.26 |
| RNN-id | 85.98 | 86.13 |
| LSTM | 90.01 | 89.86 |
| LSTM-chrono | 88.10 | 88.43 |
| GRU | 92.16 | 92.39 |
| JANET | 92.50 | 91.94 |
| SRU | 92.79 | 92.49 |
| GORU | 86.90 | 87.00 |
| NRU | 95.46 | 95.38 |
| Phased LSTM | 88.76 | 89.61 |
| LMU | **96.97** | **97.15** |
| FF-baseline | 92.37 | 92.65 |

**Discussion**  The LMU surpasses state-of-the-art by achieving 97.15% test accuracy, despite using only 468 internal state-variables and ~102k parameters. We make three important observations regarding the LMU's performance on this task: (1) it learns quickly, exceeding state-of-the-art in 10 epochs (the results from Chandar et al. [7] use 100 epochs for comparison); (2) it is doing more than simply memorizing the input (by outperforming the baseline it must be leveraging the hidden nonlinearities to perform some useful computations across the memory); and (3) since $d = 256$ is significantly less than 784, and the input sequence is highly discontinuous in time, it must necessarily be learning a strategy for writing features to the memory cell (equation 7) that minimizes the information loss from compression of the window onto the Legendre polynomials.

### 3.3 Mackey-Glass Prediction

The Mackey-Glass (MG) data set [27] is a time-series prediction task that tests the ability of a network to model chaotic dynamical systems. MG is commonly used to evaluate the nonlinear dynamical

processing of reservoir computers [17]. In this task, a sequence of one-dimensional observations—generated by solving the MG differential equations—are streamed as input, and the network is tasked with predicting the next value in the sequence. We use a parameterized version of the data set from the Deep Learning Summer School (2015) held in Montreal, where we predict 15 time-steps into the future with an MG time-constant of 17 steps.

**Task Difficulty**  Due to the "butterfly effect" in chaotic strange attractors (i.e., the effect that arbitrarily small perturbations to the state cause future trajectories to exponentially diverge), this task is both theoretically and practically challenging. Essentially, the network must use its observations to estimate the underlying dynamical state of the attractor, and then internally simulate its dynamics forward some number of steps. Takens' theorem [35] guarantees that this can be accomplished by representing a window of the input sequence and then applying a static nonlinear transformation to this delay embedding. Nevertheless, since any perturbations to the estimate of the underlying state diverge exponentially over time (at a rate given by its Lyapunov exponent), the time-horizon of predictions with bounded-error scales only logarithmically with the precision of the observer [33].

**Model Specification**  We compare three architectures: one using LSTMs; one using LMUs; and a hybrid that is half LSTMs and LMUs in alternating layers. Each model stacks 4 layers and contains ~18k parameters. To balance the number of parameters, each LSTM layer contains 25 units, while each LMU layer contains $n = 49$ units and $d = 4$ memory dimensions. We set $\theta = 4$ time-steps, and did not try any other values of $\theta$ or $d$. All other settings are kept as their defaults. Lastly, since our LMU lacks any explicit gating mechanisms, we evaluated a hybrid approach that interleaves two LMU layers of 40 units with two LSTM layers of 25 units.

**Evaluation Metric**  We report the normalized root mean squared error (NRMSE):

$$\sqrt{\frac{\mathbb{E}\left[(Y - \hat{Y})^2\right]}{\mathbb{E}\left[Y^2\right]}} \qquad (8)$$

where $Y$ is the ideal target and $\hat{Y}$ is the prediction, such that a baseline solution that always predicts 0 obtains an NRMSE of 1. For this data set, the identity function (predicting the future output to be equal to the input) obtains an NRMSE of ~1.623.

Table 2: Mackey-Glass results

| Model | Test NRMSE | Training Time (s/epoch) |
|-------|------------|-------------------------|
| LSTM  | 0.079      | 20.34 s                 |
| LMU   | 0.054      | **12.89** s             |
| Hybrid| **0.050**  | 16.21 s                 |

# 4   Characteristics of the LMU

**Linear-Nonlinear Processing**  Linear units maximize the information capacity of dynamical systems, while nonlinearities are required to compute useful functions across this information [19]. The LMU formalizes this linear-nonlinear trade-off by decoupling the functional role of $d$ linear memory units from that of $n$ nonlinear hidden units, and then using backpropagation to learn their coupling.

**Parameter-State Trade-offs**  One can increase $d$ to improve the linear memory ($\mathbf{m}$) capacity at the cost of a linear increase in the size of the encoding parameters, or, increase $n$ to improve the complexity of nonlinear interactions with the memory ($\mathbf{h}$) at the expense of a quadratic increase in the size of the recurrent kernels. Thus, $d$ and $n$ can be set independently to trade storage for parameters while balancing linear memory capacity with hidden nonlinear processing.

**Optimality and Uniqueness**  The memory cell is optimal in the sense of being derived from the Padé [30] approximants of the delay line expanded about the zeroth frequency [38]. These approximants have been proven optimal for this purpose. Moreover, the phase space of the memory maps onto the unique set of orthogonal polynomials over $[0, \theta]$ (the shifted Legendre polynomials) up to a constant scaling factor [24]. Thus the LMU is provably optimal with respect to its continuous-time memory capacity, which provides a nontrivial starting point for backpropagation. To validate this characteristic, we reran the psMNIST benchmark with the diagonals of $\mathbf{A}$ perturbed by $\epsilon \in \{-0.01, -0.001, 0.001, 0.01\}$. Despite retraining the network for each $\epsilon$, this achieved sub-optimal test performance in each case, and resulted in chance-level performance for larger $|\epsilon|$.

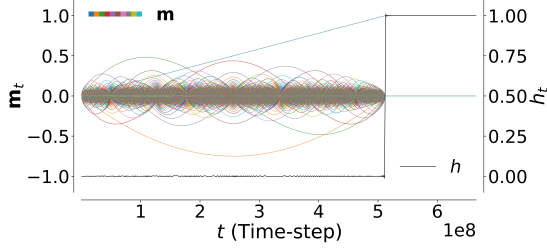
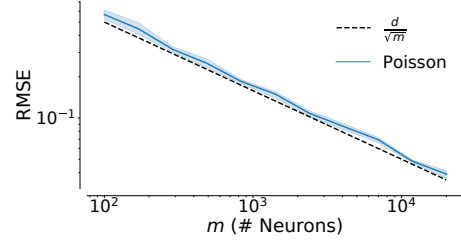

Figure 4: LMU memory ($d = 10{,}240$) given $u_t = 1$.    Figure 5: $\mathcal{O}(d/\sqrt{m})$ scaling.

**Scalability**    Equations 1 and 2 have been scaled to $d = 10{,}240$ to accurately maintain information across $\theta = 512{,}000{,}000$ time-steps, as shown in Figure 4 [37]. This implements the dynamical system using $\mathcal{O}(d)$ time and memory, by exploiting the structure of $(\mathbf{A}, \mathbf{B})$ as shown in Figure 6. We find that the most difficult sequences to remember are pure white noise signals, which requires $\mathcal{O}(d)$ dimensions to accurately maintain a window of $d$ time-steps.

## 5   Spiking Implementation

The LMU can be implemented with a spiking neural network [38], and on neuromorphic hardware [28], while consuming several orders less energy than traditional computing architectures [6, 37]. Here we review these findings and their implications for neuromorphic deep learning.

The challenge in this section pertains to the substitution of static nonlinearities that emit multi-bit activities every time-step (i.e., "rate" neurons) with spiking neurons that emit temporally sparse 1-bit events. We consider Poisson neurons since they are stateless, and thus serve as an inexpensive mechanism for sparsifying signals over time – but this is not a strict requirement. More generally, by reducing the amount of communication and converting weight multiplies into additions, spikes can trade precision for energy-efficiency on neuromorphic hardware [6, 12]. Moreover, this can be accomplished while preserving the optimizations afforded by deep learning [31].

**Neural Precision**    The dynamical system for the memory cell can be implemented by mapping each state-variable onto the postsynaptic currents of $d$ individual populations of $p$ Poisson spiking neurons with fixed heterogeneous tuning curves [14, 38]. We consider the error between the ideal input to the original rate neuron representing some dimension, versus the weighted summation of spike events representing the same dimension. Theorem 3.2.1 from [37] proves that this error has a variance of $\mathcal{O}(1/p)$. By the variance sum law, repeating this for $d$ independent populations yields an overall RMSE of $\mathcal{O}(\sqrt{d/p})$. Letting $m = pd$ be the total number of neurons, we find that the error scales as $\mathcal{O}(d/\sqrt{m})$, as validated in Figure 5. This grants access to a free parameter that trades precision for energy-efficiency, while scaling to the original network in the limit of large $m$ [37].

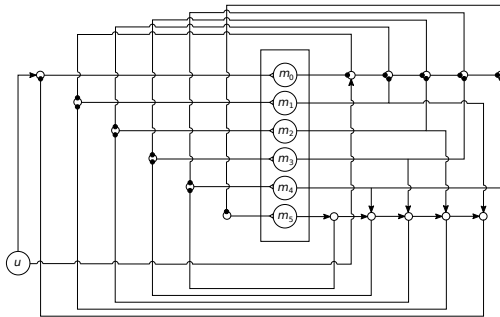

Figure 6: Connection structure ($d = 6$) adapted from [37]. Forward arrow heads indicate addition, circular heads indicate subtraction. The $i^{\text{th}}$ state-variable continuously integrates its input with a gain of $(2i + 1)\,\theta^{-1}$.

**Neuromorphic Implementation**    This spiking neural network has been implemented on neuromorphic hardware including Braindrop [28] and Loihi [37]. Each population is coupled to one another to implement equation 1 by converting the postsynaptic filters into integrators [38]. This results in a specific connectivity pattern, shown in Figure 6, that exploits the alternating structure of equation 2. An ideal implementation of this system requires $m$ nonlinearities, $\mathcal{O}(m)$ additions, and $d$ state-variables. Spiking neurons may also be used to implement the hidden state of the LMU by nonlinearly encoding the memory vector [38]. Since this scales linearly in time and memory, with `sqrt` precision, the LMU offers a promising architecture for low-power RNNs.

# 6 Discussion

Advanced regularization techniques, such as recurrent batch normalization [11] and Zoneout [21] are able to improve standard RNNs to perform near state-of-the-art on the recent psMNIST benchmark (95.9%). Without relying on such techniques, our recurrent architecture surpasses state-of-the-art by a full percent (97.15%), while using fewer internal units and state-variables (468) than image pixels (784). Nevertheless, recurrent batch normalization and Zoneout are both fully compatible with our architecture, and the potential benefits should be explored.

To our knowledge, the LMU is the first recurrent architecture capable of handling temporal dependencies across 100,000 time-steps. Its strong performance is attributed to the cell structure being derived from first principles to project continuous-time signals onto $d$ orthogonal dimensions. The mathematical derivation of the LMU is critical for its success. Specifically, we find that the LMU's dynamical system, when coupled with a nonlinear function, endows the RNN with several non-trivial advantages in terms of learning long-range dependencies, training quickly, and representing task-relevant information within sequences whose length exceeds the size of the network.

The LMU is a rare example of deriving RNN dynamics from first principles to have some desired characteristics, showing that neural activity is consistent with such a derivation, and demonstrating state-of-the-art performance on a machine learning task. As such, it serves as a reminder of the value in pursuing diverse perspectives on neural computation and combining tools in mathematics, signal processing, and deep learning.

The basic design of our layer, which consists of a nonlinear hidden state and linear memory cell, presents several opportunities for extension. Preliminary work in this direction has indicated that introducing an input gate is beneficial for problems that require latching onto individual values (as required by the adding task [18]). Likewise, a forget gate yields improved performance on problems where it is helpful to selectively reset the memory (such as language modelling). Lastly, we have proposed a single memory cell per layer, but in theory one can have multiple independent memory cells, each coupled to the same hidden state with a different set of encoders and kernels. Multiple memories would enable the same hidden units to write multiple streams in parallel, each along different time-scales, and compute across all of them simultaneously.

**Acknowledgments**

We thank the reviewers for improving our work by identifying areas in need of clarification and suggesting additional points of validation. This work was supported by CFI and OIT infrastructure funding, the Canada Research Chairs program, NSERC Discovery grant 261453, ONR grant N000141310419, AFOSR grant FA8655-13-1-3084, OGS, and NSERC CGS-D.

## Footnotes

[1] https://github.com/abr/neurips2019

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
