[Reviews · NeurIPS 2019]

Reviewer 1



Originality: the use use of the Legendre polynomial seems rather creative, it was certainly important to define RNNs with good models of coupled linear units. Quality: The set of benchmarks is well chosen to describe a broad scope of qualities that RNN require. One non-artificial task would have been a plus though. What would have been even more important is to support the theory by controlling the importance of the initialization of the Matrices A and B. What if A was initialized with a clever diagonal (for instance the diagonal of A_bar)? As the architecture is already rather close to the one of NRU, one may wonder whether the architecture is not doing most of the job. On a similar topic, the authors say that A and B may or may not be learned. I might find it useful to know for which tasks are the matrices A, B and W_m being learned? Clarify: The paper is very well written. Only the section on Spiking neural network could be a little better described. I do not find it clear, what is the challenge discussed in the neural precision section? What is the "error" mentioned line 268? I understood that this spiking neural network only models the linear part of the LMU and not the non-linear units, but I remain unsure because of line 284. Why in this linear scaling (l. 285 to 289) particularly good? in comparison to what is that good? I would assume that a dedicated digital hardware transmitting floating numbers but exploiting this clever connectivity would be much more accurate than the noisy Poisson network and avoid using p neurons per memory unit. ---- Second Review ----- Thanks to the authors for giving numerical results with other choices of A_bar. I think the paper would greatly benefit from the insertion of one or two sentences about that to prove that the mathematical depth the paper is not a decoration but leads to untrivial improvements on machine learning benchmarks. However, I also find it hard to believe that the diagonal of A_bar alone leads to chance level on permuted sequential MNIST. I am not satisfied with the justification of the authors that this is due to a bad parameter initialization and I think it would deserve a bit more analysis to either give a more convincing reason or find the trick to make A_bar diagonal work - at least a bit - in this setup. It looks like this naive change of A_bar is ill-defined if it achieves chance level. I think that after some rewriting efforts on the spiking hardware section to clarify what is done and the relevance of that approach, the paper will become an excellent paper.

Reviewer 2



This paper clearly differentiates its novelty from the work on which it builds. The quality of the paper is good. The paper is mostly clear; but the abstract could be made clearer by focusing more at the beginning on the impact of the LMU than on how it works. Also, throughout the paper I was wondering whether this work was all simulation or whether it was run on real LMU hardware. The paper mentions BrainDrop and Loithi multiple times, and Section 5 suggests a hardware implementation too; but in the end, I decide everything was simulated because you pointed to code in GitHub. But you didn't say whether that code needed special hardware to run. So which is correct? It would help to clarify that early on in the paper. As for significance, this work will be of interest to the neural chip researchers at NeurIPS. I definitely enjoyed that the authors were able to use references from 1892 and 1782! :-)

Reviewer 3



I read the other reviews and the author feedback addresses the raised questions / concerns, in particular about the dimensionality of u and comparison to phased LSTM. I think this paper addresses a relevant problem (limited memory horizon), presents a novel approach (LMU), and nicely analyses this approach theoretically and empirically. I raise my score to 8. Originality: Even though the proposed approach is based on the work "Improving Spiking Dynamical Networks: Accurate Delays, Higher-Order Synapses, and Time Cells", its transfer to deep neural networks is new and of significance. In the related work, "Phased LSTM: Accelerating Recurrent Network Training for Long or Event-based Sequences" could be discussed and also used as a baseline. Quality: The proposed approach is well described and seems easy to implement from the given description in the text. The provided theoretical formulation for the continuous-time case yields a valuable interpretation / intuition and experiments are well tailored to show-case the strengths of LMU module, in particular the increased memory horizon. Also chapters 4 & 5 describing the characteristics of the LMU and a spiking variant are helpful and provide a summary of interesting features. Clarity: The work is well written and structured. Figures and tables are of proper quality. It seems, that one could easily reproduce the results with the provided code and descriptions.

[Author Response · NeurIPS 2019]

**Reviewer #1**   In response to the suggestion to control for initialization of $\mathbf{A}$ and $\mathbf{B}$, we explored several alternatives to $\bar{\mathbf{A}}$, including: (a) the identity matrix, (b) the all-zero matrix, (c) the identity shifted down one row (i.e., a queue of length $d$), (d) only the diagonals of $\bar{\mathbf{A}}$, (e) only the *off*-diagonals of $\bar{\mathbf{A}}$, and (f) the original $\bar{\mathbf{A}}$ with its diagonals perturbed by some small $\epsilon$. We kept all else consistent with the original setup (i.e., no training on $\bar{\mathbf{A}}$ and $\bar{\mathbf{B}}$) and retrained the model for each alternative. We found that (c) achieved 87.58% test accuracy, consistent with RNN results from Chandar et al. Performance for (f) was a cusp centered about $\epsilon = 0$. Although we only had time to collect a few datapoints, we obtained 94.73% for $\epsilon = -0.01$, 96.97% for $\epsilon = -0.001$, and 96.59% for $\epsilon = +0.001$ (compared to 97.15% for $\epsilon = 0$). All other choices (a, b, d, e), as well as larger values of $|\epsilon|$, resulted in chance-level performance (~10%) on every epoch – indicating poorly initialized weights. Setting $\bar{\mathbf{A}}, \bar{\mathbf{B}}$ to be trainable did not help. These results empirically support the theory that the LMU's linear dynamical memory is critical for its success.

We have observed that further tuning of $\bar{\mathbf{A}}, \bar{\mathbf{B}}$, and $\theta$ is unnecessary for these tasks (lines 103–104) and often counter-productive. However, training these parameters improved performance for small $\theta$ in the memory capacity task.

The challenge in the neural precision section relates to substituting "rate" neurons that emit 64-bit floating point values every time-step, with spiking neurons that emit temporally sparse 1-bit values (i.e., binary events that are mostly zero). The error mentioned on line 268 is the RMSE between the ideal input to a rate neuron representing a dimension, and the weighted summation of spike events representing that dimension. This scaling is notable because it converges to zero as the number of neurons are increased (this is nontrivially shown by the referenced theorem). This guarantees that, given enough neurons, there is no systematic error that could prevent the system from approaching the ideal. The Poisson neuron was highlighted since it is entirely *memoryless*. For this, or for other neuron models, spikes provide a trade-off between energy (e.g., in neuromorphic hardware) and model accuracy (Blouw et al., 2019). Lastly, we note that Voelker and Eliasmith (2018) show that spiking neural nonlinearities perform a similar role to the hidden nonlinear units in the LMU. We hope to include these clarifications in a rewrite.

**Reviewer #2**   All experiments were run on GPUs using Keras and TensorFlow. We will clarify this early in the paper and focus on the impact of the LMU earlier in the abstract, in a rewrite. Our GitHub page will mention hardware specs, instructions for running on GPUs or CPUs, and provide examples of how to use the LMU within Keras. We view our work as an important step to realizing the performance benefits of RNNs on low-power spiking neuromorphic chips, which historically have been eclipsed by the accuracy and parallelism of GPUs. To support this, the code for the neuromorphic implementations will also be made available.

Equation 7 was motivated by the need to project all potentially relevant information into a single memory, and is similar to the design of the NRU (line 81). We resume this discussion with reviewer #3. We believe that Figure 3 demonstrates the potential for this theory to scale (note the `1e8` on the $t$-axis). In this case showing that we can process an input sequence containing a billion elements using limited resources.

**Reviewer #3**   Phased LSTMs address similar challenges by way of a novel "time gate" designed to filter updates at particular frequencies. By using the `phased-lstm-keras` package, and an otherwise identical experimental setup, we achieved 83.63% test accuracy on psMNIST with phased LSTMs. The model is sized to ~102k parameters, uses the default leak on the time gate ($\alpha = 0.001$), and is consistent with the sequential (non-permuted) MNIST example included in their repository. Reducing $\alpha$ by 10x, which makes it closer to a vanilla LSTM, and using ~165k parameters, improved accuracy to 89.61% (c.f., vanilla LSTM $\implies$ 89.86%). Setting $\alpha = 0$ provided no further improvements. This suggests that although phasing improves LSTMs for the sequential MNIST task, it does not appear to help for the permuted task variant wherein the input sequences no longer contain prominent cyclic features.

Equations 4, 6, and 7 are much closer to the referenced NRU than the LSTM in terms of the design of the module. Only when $d = 1$ does the layer becomes somewhat analogous to a single-unit LSTM, in that its feedback dynamics behave as a controllable leaky integrator for the hidden state. For larger $d$, a distinguishing feature is the particular choice of $\bar{\mathbf{A}}$ and $\bar{\mathbf{B}}$ which couple linear units to optimally represent a continuously sliding window of $u_t$. Equation 3 establishes a mathematical link between the Legendre polynomials, the memory vector ($\mathbf{m}$), and the history of $u_t$.

Multiple dimensions are taken into account by way of the encoding vectors ($\mathbf{e_x}$, $\mathbf{e_h}$, and $\mathbf{e_m}$), which project across all dimensions. But as noted, $u_t$ is one-dimensional, since there is exactly one memory vector in a given layer. Extensions that use multiple memories (each with its own encoders and memory kernel) will be considered in future work (lines 310–311). This would enable the network to maintain multiple windows in parallel within the same layer.

The choice of $d$, together with the frequency content of the window, determines the adversarial sequences. In the continuous-time domain, polynomial windows of degree $\gg d$ are adversarial (by equation 3 and orthogonality of Legendre polynomials). In the discrete-time domain, the most difficult sequences are those with power at the Nyquist frequency. For example, we find that both Gaussian noise and Brownian motion require approximately $2d$ dimensions in order to accurately maintain a window of $d$ time-steps. Further mathematical analysis is required.

[Meta-Review · NeurIPS 2019]

This paper proposes a new memory layout for recurrent neural networks that is 1. theoretically grounded 2. allows for orders of magnitude longer memory than traditional approaches with comparable parameter cost The results are also confirmed experimentally. This work is definitely of interest to Neurips community and would be a great contribution to the conference.